# Two-View Structure-from-Motion with Multiple Feature Detector Operators

**Elisabeth Johanna Dippold [1] and Fuan Tsai [1,2,*]**

1. Department of Civil Engineering, National Central University, 300, Zhongda Rd., Zhongli, Taoyuan 32001, Taiwan
2. Center for Space and Remote Sensing Research, National Central University, 300, Zhongda Rd., Zhongli, Taoyuan 32001, Taiwan
* Correspondence: ftsai@csrsr.ncu.edu.tw

**Abstract:** This paper presents a novel two-view Structure-from-Motion (SfM) algorithm with the application of multiple Feature Detector Operators (FDO). The key of this study is the implementation of multiple FDOs into a two-view SfM algorithm. The two-view SfM algorithm workflow can be divided into three general steps: feature detection and matching, pose estimation and point cloud (PCL) generation. The experimental results, the quantitative analyses and a comparison with existing algorithms demonstrate that the implementation of multiple FDOs can effectively improve the performance of a two-view SfM algorithm. Firstly, in the Oxford test dataset, the RMSE reaches on average 0.11 m (UBC), 0.36 m (bikes), 0.52 m (trees) and 0.37 m (Leuven). This proves that illumination changes, blurring and JPEG compression can be handled satisfactorily. Secondly, in the EPFL dataset, the number of features lost in the processes is 21% with a total PCL of 27,673 pt, and this is only minimally higher than ORB (20.91%) with a PCL of 10,266 pt. Finally, the verification process with a real-world unmanned aerial vehicle (UAV) shows that the point cloud is denser around the edges, the corners and the target, and the process speed is much faster than existing algorithms. Overall, the framework proposed in this study has been proven a viable alternative to a classical procedure, in terms of performance, efficiency and simplicity.

**Keywords:** two-view Structure-from-Motion; point cloud generation; SURF; FAST; ORB

## 1. Introduction

The estimation of a 3D structure from two images can be beneficial for the generation of Digital Surface Models (DSM), as well as for navigation, urban planning or object model reconstruction tasks. The estimation of a 3D structure with epipolar geometry needs two or more frames with overlapping content, as well as additional information about the sensor. Through feature detection, feature matching and view generation, the 3D structure can be reconstructed. Incorporating three Feature Detector Operators into a two-view Structure-from-Motion algorithm can further enhance the performance of the feature detection step and improve the overall 3D reconstruction performance. The enhancement of 3D reconstruction while preserving or improving motion estimation can be challenging. The range sensed and the number of images, as well as additional data, are crucial to tackling this challenge.

Global range or space-based remote sensing provides observation of land, atmosphere and ocean [1]. In contrast, close-range digital photogrammetry provides observations of a target close to the sensor. Similar to Computer Vision (CV), the tasks in photogrammetry include the gain of information to 3D reconstruction of the scene from a set or pair of 2D images. Where the sensor was and what was captured or sensed [2] is the essential question. A possible solution is a Structure-from-Motion (SfM) algorithm that estimates pose (motion) and obtains structure (3D point cloud).

The framework of SfM algorithms is a combination of image processing techniques used to chase the ray. This includes the extraction and description of the image data and the sensor-dependent information [3]. Some frameworks are striving to enhance the image quality [4], to analyze the input/output and structure [5], camera model [6] and compression [7].

The acquired set of data can relate to the sensor, among others, as mono—one sensor [8,9] or stereo—two sensors [10,11]. Further, the size of the used dataset can vary for both mono and stereo, so that single view [12,13], two-view [14–16] and multi-view [17,18] SfM algorithms are in use. In addition, the data acquisition technique to obtain data can sense from various distances (target to sensor), for instance, application of satellite imagery [19], UAV [20], and close range [21]. In addition, a dataset can be sensed under lab conditions [22], computer-generated/changed [23] or real-world (unchanged) conditions [24].

The overall workflow for an SfM algorithm can be divided into two phases. Firstly, the correspondence which consists of feature detection, extraction, description and matching, as well as geometric verification. Secondly, the actual process to obtain a 3D reconstruction by image registration, triangulation and bundle adjustment [25–27]. The choice of an underlying strategy and how to process the image pairs are various. Among others, they can be based on an incremental [25], a hierarchical [28] or a global [29] approach.

The establishment of an image relation to prepare the feature matching is based on detected features, which are area-based or relational. Relational matching detects and describes geometric or other patterns, whereas the area-based approach takes the whole image into account to detect patterns based on the intensity values like Grey-Level-Cooccurrence Matrix (GLCM). In contrast, a feature-based process proceeds pixel-wise in order to detect key points, edges and similar smaller units [10]. Several versatile studies have been conducted around the topic of feature detection and matching, i.e., a benchmark, different sensors or applications including a benchmark for consumer cameras datasets to evaluate local features concerning the application [30]; and the choice of a feature detector operator for seasonal change in satellite imagery [31]. The performance of feature detection and matching is relevant for localization and mapping only. The feature-based approach, as described, works best with a large choice of different algorithms and the performance difference according to the use case can vary.

This study focuses, among others, on clear structures like corners and edges [32]. A corner or the intersection of two edges is a stable, and based on the intensity values, a robust feature. A classical corner detector is Harris Corner Detector (HCD) [33]. FAST, Features from Accelerated Segmented Test [34], is a more computationally efficient corner detector based on HCD. In addition, the extended version of FAST and HCD, is Oriented FAST and Rotated BRIEF (ORB) [35]. ORB can handle multiple scales so that it can deal with rotation and invariance. SURF (Speeded Up Robust Features) [36], similarly to FAST and ORB, is based on the Hessian matrix utilized in HCD. These three feature detectors share the same fundamentals (from HCD). As a result, the combination of different detectors which share the same origin of HCD seems to be reasonable. In addition, this idea is similar to the development of HCD, which is a combination of corner and edge detection.

The number and quality of features detected to generate a dense point cloud depend on several tasks. Firstly, the determination of the camera poses of the first two frames is processed. Secondly, the ratio of good matches in contrast to miss matches is considered. Further, the ability of the algorithm to deal with a small number of features, including weak features, and also the ability to separate good matches from missed matches [28,29]. Finally, the design of a general-purpose workflow. There are four main challenges in processing (un)ordered images to generate a 3D Model: robustness, accuracy, completeness and scalability [37].

The evaluation of a point cloud can be realized among others in two ways. Firstly, the neighborhood calculation that displays the geometric features of a point related to its surroundings, i.e., density-based contour extraction. Secondly, geometric features are

displayed with respect to height, plane or Eigenvalues of the point. Height features are obtained by taking the difference between height models like DSM and DEM (Digital Surface Model; Digital Elevation Model). The evaluation of a plane includes the fitting of a surface to a plane by distance. The feature evaluation based on Eigenvalues and Eigenvectors opens up a big catalogue of opportunities, for instance, for planar and nonplanar areas, as well as linear edges [38,39].

This study aims to extend its application to a wider PCL generation and validity, as well as better performance and reducing required computation. In particular, maintaining the number and quality of matches while increasing the number of feature points improves the 3D reconstruction efficiently. The selection of the applied FDO targets aims to strengthen corner and edge properties of the final 3D model. The implementation of multiple FDOs into a two-view SfM algorithm tackles the crucial first step of establishing a robust and reliable initial view. Experimental results demonstrate that the applications with close-range photogrammetry as well as with UAV datasets both produce satisfactory outcomes. The rest of this study is organized as follows: Sections 2–6, References.

## 2. Related Work

Structure-from-Motion (SfM) is characterized mainly by processing images with respect to their camera positions. The SfM algorithm investigates the possibility to reconstruct 3D models from 2D images. There are three fundamental steps in SfM, including (I) feature detection and extraction of points [35,36], lines [40] or other key elements such as corners [41] to establish a relation between the images; (II) pose estimation of the camera sensor movement [42–44] to guide the triangulation [45]; and (III) the actual 3D reconstruction based on features and camera motion to obtain the final output, the point cloud [46].

In addition, artificial intelligence (AI) technology is also an efficient and cost-effective approach for SfM-based 3D reconstruction. The reconstruction can be modelled as point cloud, mesh or geometric models [47] in order to represent, visualize or record the reconstructed objects [48] and for subsequent applications. Machine learning-based methods have been successfully applied to areas such as 3D image to mesh generation [49], mapping to 3D model over time (4D) as multi-temporal building model [50], complex high-rise grid structures [51] and the medical field in general [52].

Although a few algorithms and software are available for SfM-based 3D reconstruction, there is still room for improvement [53]. For works focusing on the processing of the first two frames, besides classical two-view SfM [14,40], other solutions such as two-view multibody SfM [54] and AI-based methods [16,55] also achieved encouraging results. In addition, open-source photogrammetry software as in [56] and [57] is also used.

A better understanding of detector performance according to a given task or system requirement can be useful beforehand. In this regard, handcrafted detectors and descriptors in comparison with those from AI are equally reviewed with respect to sensor and image condition changes [58]. An implementation to improve a sparse point cloud generation and address the distribution was proposed in [59]. In general, feature detector operators consist of a detector and a descriptor. These can be treated as a unit to evaluate the performance [60] or as a group [61]. Alternatively, studies are utilizing fusing FAST and Harris [62], as well as combining point and line features [63].

Further, it is important to consider which FDO of the selected group is suitable for a given task focusing on processing time, accuracy, robustness, etc. In contrast, the performance evaluation of the combination of different detectors and descriptors [64] or sensitivity to a particular material [31] has been conducted in other studies. These surveys can lead to novel workflows and implementation above and beyond classical approaches, by thinking outside of the box to investigate further. The authors of this study decided to implement SURF, ORB and FAST with BRIEF. The individual analyses have been conducted [65] on the Oxford dataset [66] with recall, precision and time.

Individually, SURF and SIFT can handle rotation and zoom variation for a textured scene (bark) well. SURF and ORB perform better for a structured scene (boat). Blur seems

to be handled best for both cases (textured and structured) by ORB and FAST with BRIEF. ORB is best for the Graf scene and for the wall scene, FAST with BRIEF and MSER with SIFT perform best in case of viewpoint change. Illumination changes (Leuven) are handled best by FAST with BRIEF, whereas for JPEG compression, ORB performs best. In addition, SURF and FAST with BRIEF quickly detect most key points [65]. As a result, these three (SURF, ORB and FAST with BRIEF) seem to perform well under certain conditions.

The evaluation of point clouds can be classified into two categories, geometric features and neighborhood. Evaluation of point clouds by geometric features is based on height, plane or Eigenvalues for oblique data such as airborne [38] or single building information modeling (BIM) [67]. An alternative approach is a mixed approach (geometric features and neighborhood) to describe the desired property of a point cloud with contour by point density as well as geometric features [39]. This approach focuses on neighborhood only to carry out reconstruction quality and overall performance [68].

This study adds to a neighborhood point cloud-driven evaluation a comparison with a point cloud generated by open source. This performance check allows overall comprehensive analyses. For this reason, this study decided to run the dataset of CRSRS in the state-of-the-art open source solution for photogrammetry—MicMac [57].

The evaluation of the combination of multiple FDOs with the same foundation seems to be reasonable, novel and promising. The application of two different Benchmark datasets to evaluate performance by separating motion and reconstruction leads to a comprehensive and independent task analysis. In addition, the detection of a large number of features in a short amount of time seems to be beneficial for further processing. The 3D reconstruction of a real-world target from images collected by UAV is used to test our workflow under real-world conditions, which complements and completes the performance analyses.

## 3. Workflow and Materials

### 3.1. Workflow

The workflow summarized (Figure 1) is divided into three blocks, input, the correspondence search and incremental reconstruction [25]. The implementation follows the key concept introduced in Figure 1 and further described in the Section 4.

Three datasets are applied to carry out behavior based on changing image conditions and for reconstruction purposes. Two datasets, Oxford and EPFL, are publicly available, whereas the CSRSR dataset consists of images collected using an unmanned aerial vehicle (UAV) and is used to evaluate the performance of the algorithms developed in this study.

The first (input) block (Figure 1 marked in green), consists of two images for the two-view SfM algorithm. The second block (Figure 1 marked in orange), for feature detection and matching, includes the implementation of multiple FDOs (SURF, ORB and FAST). It is evaluated with the Affine Covariant Features dataset [66] from Oxford. The evaluation covers the RMSE of Homography matrix estimation and the number of features detected. The third (view generation) block (Figure 1 marked in blue), involves the fundamental matrix estimation, triangulation and the output of the point cloud.

The reconstruction is evaluated based on the first two images of the Fountain-P11 dataset [69] from École Polytechnique Fédérale de Lausanne (EPFL). In addition, 3D reconstruction of a real-world target uses a dataset captured by UAV from the CSRSR.

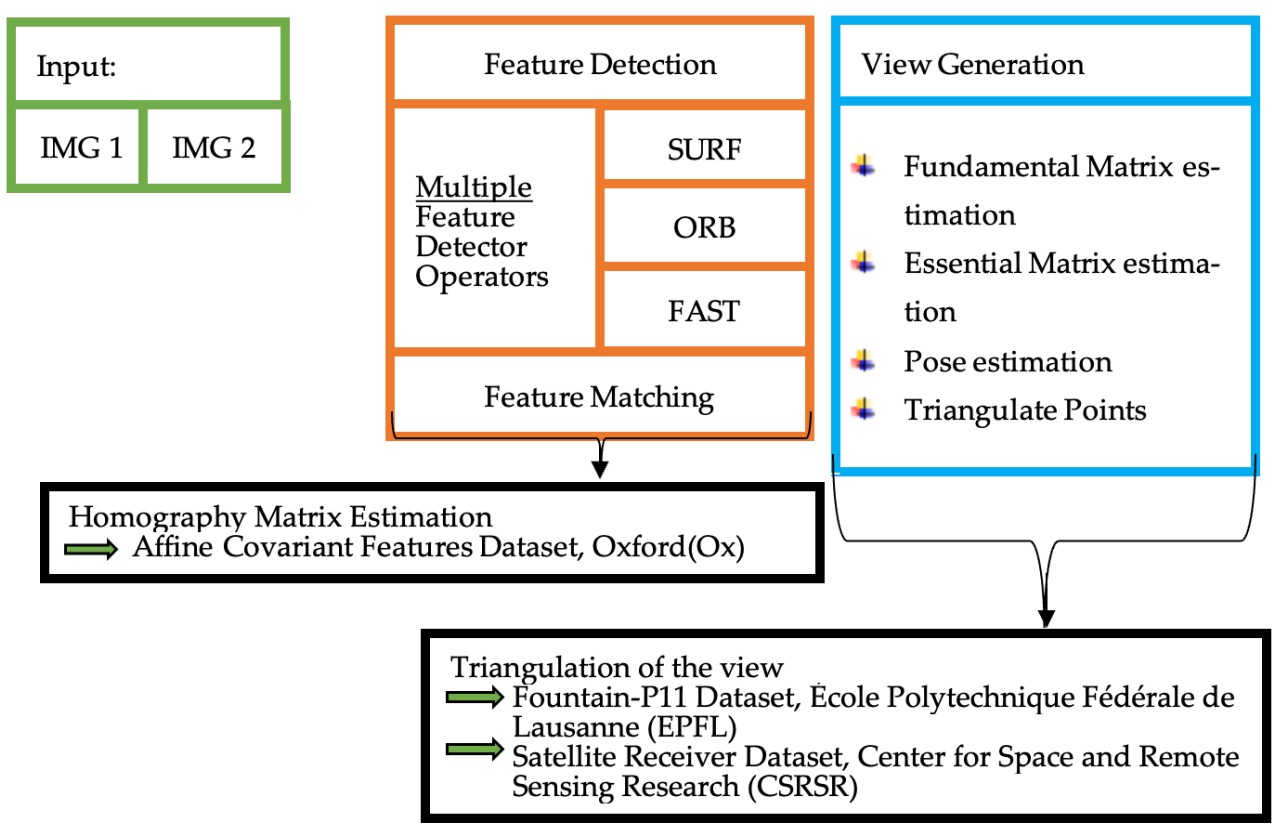

**Figure 1.** Workflow of the two-view SfM algorithm with the implementation of multiple FDOs.

### 3.2. Affine Covariant Features Datasets

The Affine Covariant Features dataset is an arbitrary image dataset to evaluate feature detectors and descriptors. The effectiveness and robustness of our proposed approach can be studied condition-wise and scene-wise. [66]. The dataset consists of eight cases with varying scene content, complexity and condition change. Every case is dedicated to one or more image condition changes. Every case contains one reference image and five follow-up images (Figure 2). The level of condition change increases over the sequence (img2–img6).

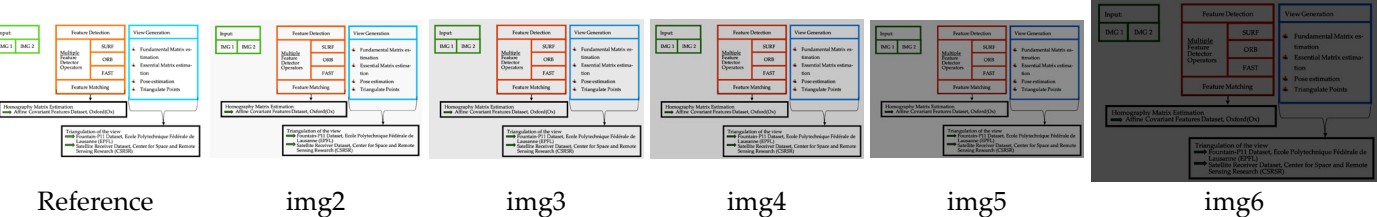

|  |  |  |  |  |  |
|---|---|---|---|---|---|
| Reference | img2 | img3 | img4 | img5 | img6 |

**Figure 2.** Example structure of image condition changes.

The dataset consists of two different scene types—textured and structured. Textured scenes are characterized by a repetitive pattern such as the bark of a tree. Structured scenes, in contrast, are a mixture of homogenous regions and clear recognizable edges and corners. Except JPEG compression and illumination, all three other changes are applied to both scene types. The illumination change over 5 images (img2–img6) is achieved by variation of the camera aperture, whereas the JPEG compression is generated by the gradual variation of the image quality from 40 to 2% in a browser window.

The application of blur, rotation, scale and viewpoint change to both scene types offers the opportunity to study the image condition changes without the influences of the scene type. The blur and scale changes are achieved by the variation of focus and zoom,

respectively, scaled up to a factor of four. Viewpoint and rotation changes range from fronto-parallel to 60 degrees, as a gradual transformation from img2 to img6.

This benchmark dataset can be evaluated by focusing on scene content, condition change and Homography matrix estimation (H-matrix). The content includes structured and textured scenes with feature-rich and homogenous areas. A numeric evaluation can be conducted based on the ground truth of the H-matrix. Thus, the RMSE of H-matrix estimation can be conducted on average and pairwise [66,70,71].

### 3.3. Fountain-P11

The Fountain-P11 is one dataset out of the dense MVS benchmark dataset collection from EPFL. The close-range dataset shows a close-up historic water fountain in a backyard. The scene combines textured and structured elements without a background. The red brick wall, similar to the Oxford wall, is a textured element. The actual historic water fountain can be considered a structured element. The connected white wall with two windows completes the corner of the backyard captured. The image shift can be described as a rotation with a small shift similar to the first image pair I12 of the Oxford dataset Graf. The first two images and the intrinsic parameters of this dataset are applied in this study to explore our two-view algorithm. A dominant target and the absence of a background characterize this scene as an experiment in a controlled environment.

The evaluation is accomplished for the reconstruction result, the number of points detected and the sparse point cloud density. In addition, visual examination and comparison of the sparse reconstruction of each FDO are performed with our approach of multiple FDOs [69].

### 3.4. CSRSR Satellite Receiver Dataset

The CSRSR Satellite Receiver dataset was generated with a UAV (Figure 3) to test a real-world application of the approach proposed in this study. The UAV used is DJI P3Pro with a focal length of 20 mm and a principal point of 1080 (pixel) in the x-direction and 1920 in the y-direction (Figure 3a). The target is the antenna shown in Figure 2b,c consisting of a dish attached to a mounting tower.

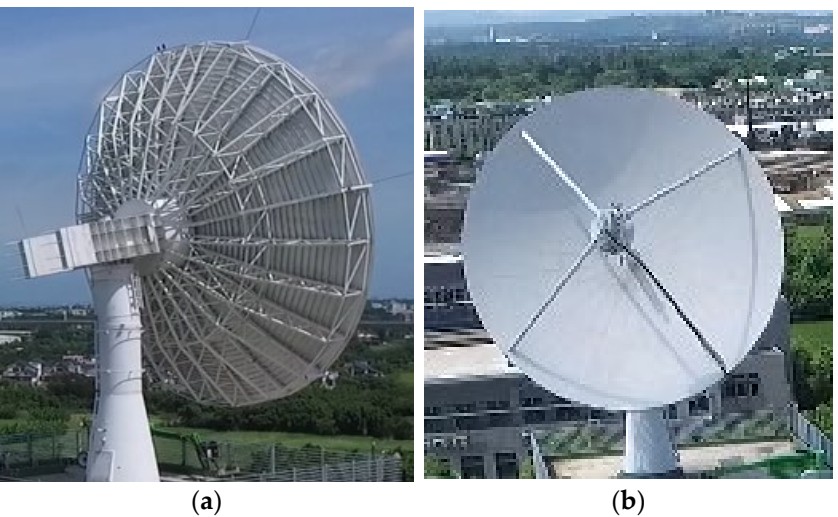

(**a**)                                        (**b**)

**Figure 3.** The antenna of CSRSR captured by UAV [72]. The back (**a**) and front (**b**) of the antenna.

The UAV imagery extracted is shown in Figure 4 and is an outdoor scene of two buildings. The image shift of the first pair (a and b) can be detected by comparing the orange marking. The change of the second pair can be seen by comparing the red square area. In general, the first pair is characterized by a small shift and angle, whereas for the second pair, the image shift and camera angle increased. This case, the first case (Figure 4a,b), can be considered as a classical close-range photogrammetry real-world

application of a two-view-based 3D reconstruction input. The second case can be described by an increased baseline and angle, in contrast to case one (Figure 4c,d). Two cases are assigned to test further capability and performance of the approach proposed in this study. The first case focuses on the antenna as shown in Figure 2b,c. The second case focuses just on the rooftop with the auxiliary build-up including the antenna.

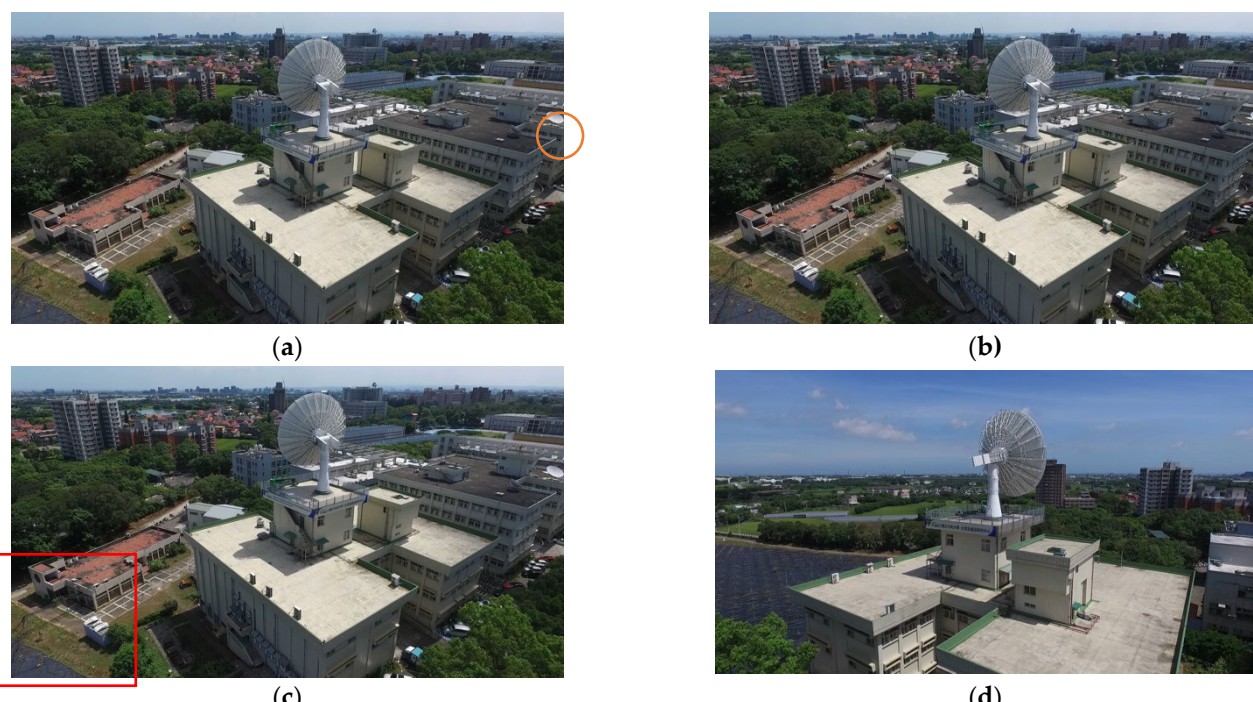

(**a**)

(**b**)

(**c**)

(**d**)

**Figure 4.** Satellite Receiver Dataset with UAV from CSRSR. The image pair (**a**,**b**) with a small shift marked by the orange circle. The image pair (**c**,**d**) with an increased motion marked with a red square.

## 4. Methodology

The key point of this study is the implementation of a multiple Feature Detector Operator (FDO), to improve the matching and reconstruction of a two-view Structure-from-Motion algorithm. The workflow of multiple FDOs into a two-view SfM algorithm is summarized in Figure 1 based on epipolar geometry (Figure 4).

Epipolar geometry extends the concept of one-view (Figure 5a) to a two-view relationship (Figure 5b). The ray of light connects the real-world P candidates with the camera, the image plane and feature point p. Two rays of light and two camera positions define a triangle, the epipolar plane. Within this triangle, the real-world point, the feature points and the cameras can be related to each other to estimate translation and rotation.

The two-view SfM limits the number of images used, as introduced in Figure 1 and as described in the Workflow and Materials section. Firstly, initialize each FDO chosen with a descriptor. The FDOs chosen in this study are all based on Harris Corner Detector (HCD)—SURF, FAST and ORB. HCD is a combined corner and edge detector. It takes the variation of the intensity value within a window. Then, it calculates the displacement of features which are stable, repeatable and with low self-similarity. That is accomplished by the detection of a target region and then distinguishing between the target and background. A corner can be described as a significant change of the intensity values in all directions. Afterwards, the non-maximum suppression (NMS) method is applied by requirements: space between corners and the total number of corners per tile [33].

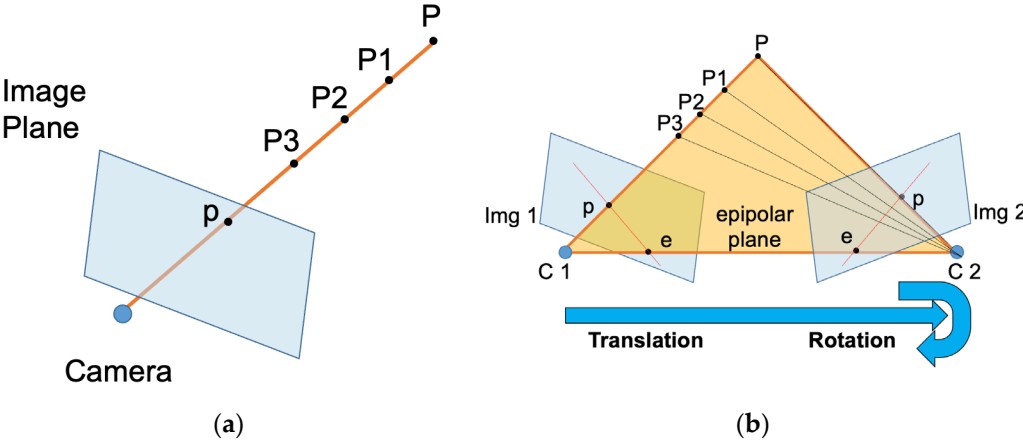

**Figure 5.** Epipolar geometry. (**a**) one view characterized by one camera position and one image capturing real world point from on point of view. (**b**) two-view characterized by two camera positions, two images capturing the same target from different looking angle.

SURF, introduced by Bay in 2006 as Speeded Up Robust Features, detects, extracts and describes feature points. The feature detection is based on the second-order Hessian matrix and the integral image, which is the result of the Gaussian second-order derivative:

$$H(x, \sigma) = \begin{bmatrix} Lxx(x, \sigma) & Lxy(x, \sigma) \\ Lxy(x, \sigma) & Lyy(x, \sigma) \end{bmatrix} \tag{1}$$

$$\det(H_{appox}) = D_{xx}D_{yy} - (0.9D_{xy})^2 \tag{2}$$

where the Hessian matrix $H(x, \sigma)$ for a point $x$ at scale $\sigma$ described in (1) and where $Lxx(x, \sigma)$ is a Gaussian second-order derivative. In addition, the approximation of $L_{xx}$, $L_{xy}$ and $L_{yy}$ is due to the box filter process as $D_{xx}$, $D_{xy}$ and $D_{yy}$ (2).

This process is sped up by a box filter to retrieve the determinant of the Hessian matrix. In order to remove weak feature points, NMS is applied, similar to HCD. The surrounding of the feature point (descriptor) is generated based on the four Haar wavelet responses [36,60].

FAST, Features from Accelerated Segment Test, introduced by Rosten and Drummond in 2006, is a feature detector (without descriptor). The default version examines the surrounding of a candidate pixel within a 7 × 7 Bresenham circle (Figure 6). If the pixels on the circle (Figure 6c, green circle—orange pixels) are brighter than p and satisfy the threshold as well, then that is a FAST corner [34,73].

BRIEF firstly smooths the path out with a randomized classification tree and with a Bayesian classifier. The response vector gets tested using a binary test between the pixels by matching candidates based on sampling geometries [74].

ORB, Oriented FAST and Rotated BRIEF, was introduced by Rublee in 2011. The combination of the FAST corner detector with the BRIEF descriptor by adding orientation property to FAST is key to ORB. FAST, as described, detects a corner by thresholding the surrounding patch (Figure 6). ORB extended this approach by adding a line connecting point and centroid, to weighting the centroid with intensity; the vector provides direction (orientation) [32]. In addition, further changes are made to achieve better rotation performance of BRIEF and become rBRIEF. Firstly, instead of deciding on one binary test, rBRIEF utilizes all binary tests for all patches. Sort the test results by distance and store them in vector T and move the first to vector R. Compare the next upcoming result stored in T with R, if threshold satisfied, copy to R; repeat that step until T is empty. At the minimum, the R vector should hold 256 tests. The greedy search results in the uncorrelated test, which means around 0.5. However, with all those changes, ORB is still sensitive to illumination changes and viewpoint differences [32,35].

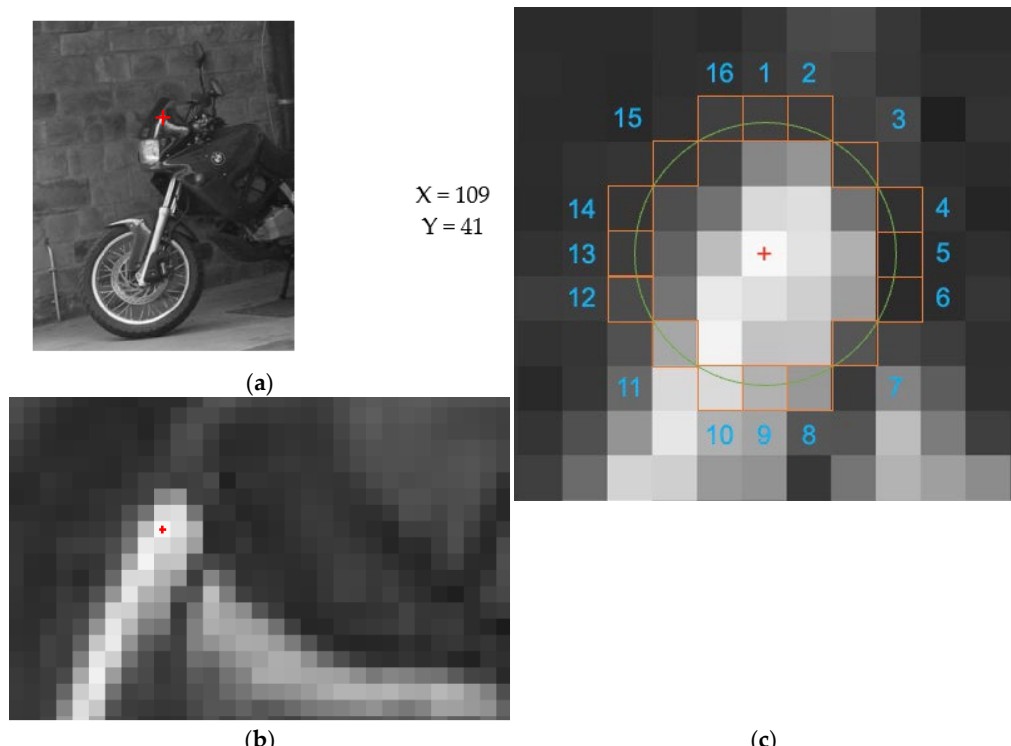

**Figure 6.** FAST. (**a**) Bike as reference image cropped (Oxford dataset), (**b**) with one corner marked (red). (**c**) Bresenham circle (green) within a 7 × 7 Patch (orange).

As described in Figure 1, the implementation parameters of the feature detection are default, except for SURF. The Hessian threshold is decreased from 400 to 30 to increase the number of features detected. The feature-matching strategy is greatly based on the data the feature descriptor provides. This study matches SURF features based on the Brute Force Norm-L2 version. The Brute Force Norm-L2 is the Euclidian norm which applies the ordinary distance (SRSS, square root of the sum of squares). ORB and FAST are matched with Brute Force Hamming distance. The distance is calculated over the number of positions until the value/symbol, etc., changes. For binary descriptors, this is a useful approach and for that reason is applied in this study for ORB and FAST. This study applies two different strategies to match the detected features of each feature detector, respectively.

The evaluation of the first part is accomplished over the RMSE of Homography Matrix (*H* matrix) estimation. The *H* matrix describes the relationship between two images. This transformation, for this study projective transformation, describes as:

$$x' = Hx \tag{3}$$

$$H = \begin{bmatrix} h_1 & h_2 & h_3 \\ h_4 & h_5 & h_6 \\ h_7 & h_8 & h_8 \end{bmatrix} \tag{4}$$

The points x of the first image can be related to the points $x'$ of the second image by Homography matrix $H$. The Homography matrix $H$ consists of nine elements; these are elementwise evaluated with the provided ground truth of the Oxford dataset with Root-Mean-Square Error (RMSE).

The combination of SURF, ORB and FAST with BRIEF and reasoning provided are provided so that the detected features can be matched. The next step of the implementation is the Homography estimation if the input is the Affine Covariant Features Dataset from Oxford (Figure 1).

The second part follows, as shown in Figure 1, the view generation, starting with the calculation of the fundamental matrix (*F* matrix), following the concept of epipolar geometry. Then, computation of the F matrix is based on eight or more points and uses an eight-point algorithm [75]. This implementation applies Random Sample Consensus (RANSAC) and adds a threshold (distance). Thus, the points far from the epipolar line are removed. The F matrix is a forerunner of the essential matrix (*E* matrix). The E matrix retrieves as follows:

$$E = (K)^T \cdot F \cdot K \tag{5}$$

where the camera matrix is *K*; the fundamental *F* calculates the essential matrix *E*. The *K* matrix consists, among others, of focal length and principal point.

The translation and rotation can be now estimated from the essential matrix and the detected feature points if the points are in front of the camera (cheirality check). The main assumption of triangulation is that the ray of light intersects at the location of the real-world point (Figure 5), so that a triangle gets defined by the real point and the camera poses. The feature points are located on the ray within the image frame (Figure 5b). The sparse point cloud, for this study, is evaluated by the point density and a visual check of the reconstruction with the Fountain dataset from EPFL and UAV dataset of CSRSR.

The final implementation generates the point cloud based on two images, the information about the principal point in x- and y-direction, and the sensors' focal length. Firstly, estimation of Fundamental and Essential Matrix is performed to describe the relation of the two images. Then, estimating the pose, which includes rotation and translation—the estimated camera movement. Finally, triangulating the features, with the knowledge of the sensors' position and movement.

## 5. Results

The evaluation is accomplished based on three datasets, two being publicly available and one being a real-world dataset. The results section is divided, for each dataset, into three parts to evaluate each dataset separately and independently.

### 5.1. Oxford Dataset

The Oxford dataset consists of ground truth (Homography matrix) for every pair. Every case, i.e., illumination (Figure 2) as explained, consists of five pairs—the reference image with a follow up in the range of img2 to img6. As introduced in Section 3, all image condition changes are applied to a structured and a textured scene, except Leuven and Ubc (Table 1). Consequently, the evaluation is group-wise structured. Firstly, bark and boat both with zoom and rotation changes. Then, bike and trees with blur changes, followed by Graf and wall with viewpoint changes. Finally, Leuven with illumination changes and Ubc with JEPG compression. The evaluation includes the RMSE pairwise presented in column bars with the absolute numbers of each group respectively. Further, in table form, the average RMSE appears on one side and the number of features detected on the other side.

**Table 1.** Oxford dataset result presentation structure.

| Name | Condition Change | Level |
|---|---|---|
| Bark | Zoom | 1 |
| | Rotation | 1 |
| Boat | Zoom | 1 |
| | Rotation | 2 |
| Bikes | Blur | 1 |
| Trees | Blur | 2 |
| Graf | Viewpoint | 1 |
| Wall | Viewpoint | 2 |
| Leuven | Illumination | - |
| Ubc | JPEG compression | - |

The first group focuses on zoom and rotation image condition changes (Table 1). The bark case is a close-up textured scene with zoom and rotation on level 1, whereas the boat scene is a structured image, which shows a boat as an equal part of a greater scene. This scene has no clear fore- and background with zoom on level 1 and rotation at level 2.

The pairwise RMSE evaluation of H-matrix estimation is shown in Figure 7. Each column is divided vertically by multi (blue), SURF (orange), ORB (gray) and FAST (yellow) and pairwise horizontally, i.e., I12—reference image 1 with image 2.

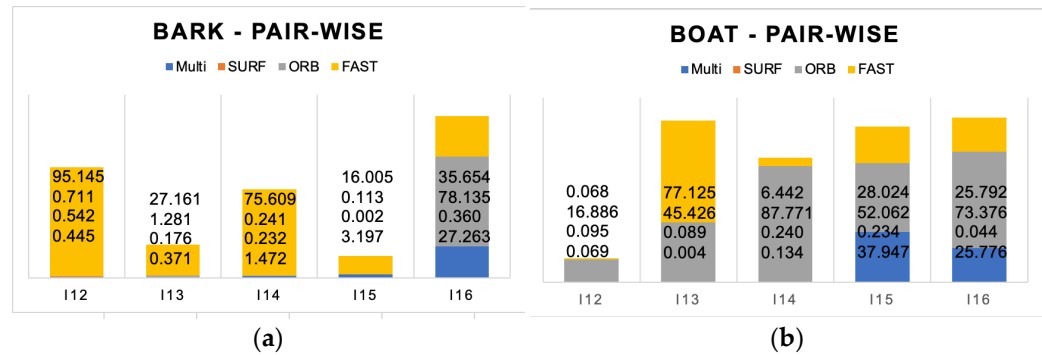

**Figure 7.** Pairwise RMSE [m] evaluation of H-matrix estimation of bark (**a**) and boat (**b**).

The textured scene, bark, seems to be most difficult for FAST even for the first pair: multi (0.445 m), SURF (0.445 m), ORB (0.711 m) and FAST (95.145 m). The multi approach of this study shows good overall performance except for the last pair. Only SURF outperforms our multi approach, except for the first pair. Until pair four (I14) of the boat case, our approach shows outstanding performance. The five (I15) and six (I16) image pairs are handled well only by SURF. The increasing zoom and rotation changes show that FAST struggles the most. However, our approach can balance the struggling operator out. FAST and ORB have higher RMSE values as the combination, multi, would imply. The structured scene (boat) seems to have better results than the textured scene. The structured scene, even with the background, adds enough intensity variety and provides enough individual features to match correctly. The textured scene, on the other hand, lacks individual features; therefore, their corrected match rate is low and RMSE is higher.

The average RMSE evaluation of H-matrix estimation and the average number of features detected for both, bark and boat, are summarized in Table 2. The orange cell marks out the best average RMSE, whereas the blue cell emphasizes the most features detected for a single operator. SURF realizes on average RMSE of 0.26 m and detects 1003 pt on the bark scene. Our multiple FDO approach achieves 6.55 m ($-6.29$ m), but detected 2700 pt ($+170\%$). For the boat scene, ORB and FAST performance was unsuitable, where SURF maintained an error under 1 m. SURF realizes an RMSE of 0.14 m and detects 2191 pt. This study of multiple FDO approaches achieves 5.96 m ($-5.82$ m), but detects 5140 pt ($+134\%$).

**Table 2.** Bark and boat average RMSE [m] and the average number of features detected.

| | AVG RMSE [m] | | Features | |
|---|---|---|---|---|
| **Column** | **1** | **2** | **3** | **4** |
| **IMG** | **Bark** | **Boat** | **Bark** | **Boat** |
| **Multi** | 6.55 | 5.96 | 2700 | 5140 |
| **SURF** | 0.26 | 0.14 | 1003 | 2191 |
| **ORB** | 16.10 | 55.10 | 124 | 224 |
| **FAST** | 49.92 | 27.49 | 1573 | 2726 |

The bike scene is a structured close-up with a dominant foreground. The image condition change applied is blur level 1, whereas the trees belong to the textured scene group and the level of blur is in contrast higher, at level 2.

The pairwise RMSE evaluation of H-matrix estimation is shown in Figure 8. The first three pairs of the bike scene show good overall performance of all FDOs under 1 m. Only SURF and our approach, multi, remain under 1 m, whereas ORB and FAST exceed 1 m. The textured scene of trees seems to retain the error under 1 m, except for one value (I14, FAST 1.292 m). The last pair, I16, shows that only ORB and FAST are able to perform well with 0.845 m and 0.817 m. Our approach is slightly higher with 1.191 m, crossing 1 m line. The blurring effect influences the sharpness of the image, so that strong features become less significant gradually. A scene with an already unclear background seems to be more beneficial than a completely sharp scene (reference image).

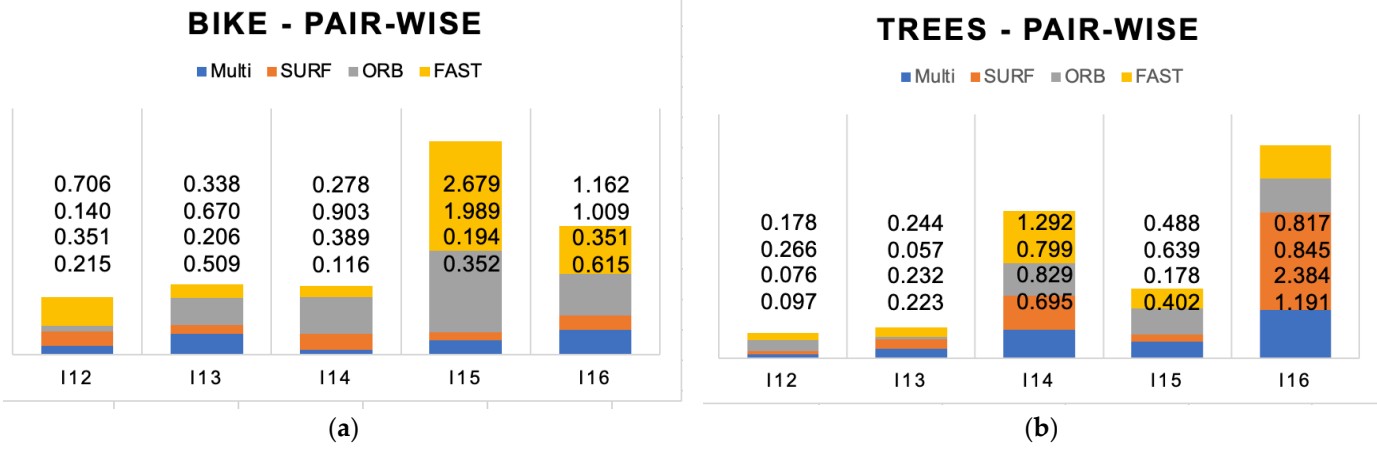

**Figure 8.** Pairwise RMSE [m] evaluation of H-matrix estimation of bike (**a**) and trees (**b**).

The average RMSE evaluation of H-matrix estimation and the average number of features detected for both, bike and trees, are summarized in Table 3. SURF can achieve an average RMSE of 0.30 m with 3462 pt of the Bikes scenes. This study's multiple FDO approach achieves 0.36 m (−0.06 m) while detecting 7907 pt (+128%) more points. The result of the trees case shows that ORB and this study's multi-FDO can both achieve 0.52 m (−+0). However, ORB detects 179 pt whereas multi-FDO detects 14,941 pt, +9833% more points.

**Table 3.** Bike and trees average RMSE [m] and the average number of features detected.

|  | AVG RMSE [m] | | Features | |
| --- | --- | --- | --- | --- |
| **Column** | **1** | **2** | **3** | **4** |
| **IMG** | **Bikes** | **Trees** | **Bikes** | **Trees** |
| **Multi** | 0.36 | 0.52 | 17,907 | 17,601 |
| **SURF** | 0.30 | 0.74 | 3462 | 2481 |
| **ORB** | 0.94 | 0.52 | 281 | 179 |
| **FAST** | 0.86 | 0.60 | 4164 | 14,941 |

The Graf and wall scenes are both textured scenes with a viewpoint change. The Graf scene is graffiti sprayed onto a flat wall; the viewpoint changes at level 1. The wall scene in contrast is an old-school brick wall as a part of a fence, with a viewpoint change of level 2. The overall performance of the first two pairs is satisfying (Figure 9). The fourth pair is difficult for our approach and that of FAST. For the fifth pair, FAST performs best; however, for I16, all FDOs are failing. The wall in contrast shows that all FDOs perform for the first four consistently well. The last pair, I16, is the most challenging one for our approach with less than 12 m error best. Viewpoint changes can be considered as the most common

challenge in real-world photography. The viewpoint changes with the position changes of the camera, translation, rotation, tilting, etc. The structured Graf scene is therefore much more challenging, even on level 1. The pattern of the graffiti seems to be more challenging than the brick wall, too.

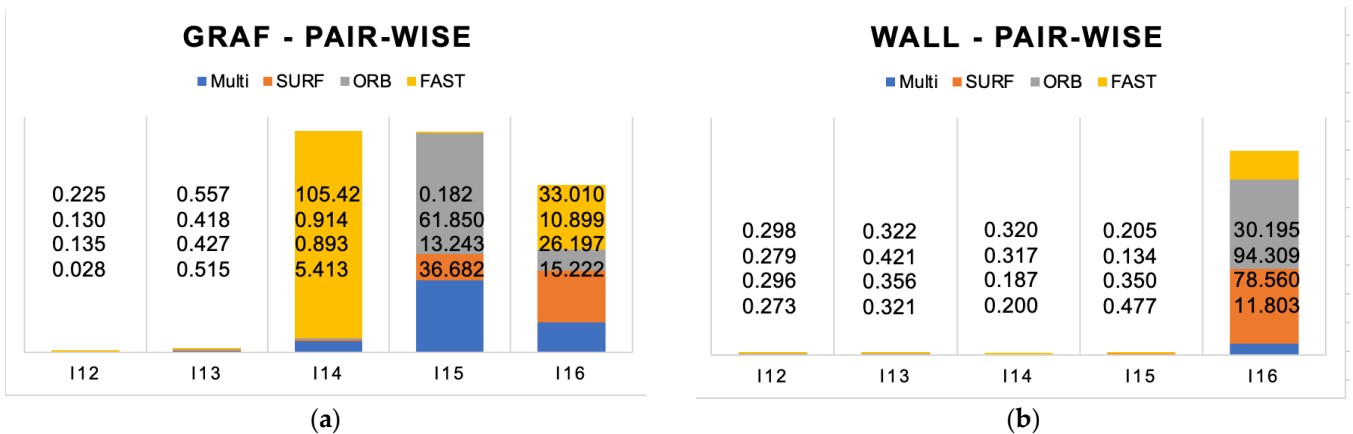

**Figure 9.** Pairwise RMSE [m] evaluation of H-matrix estimation of Graf (**a**) and wall (**b**).

The average RMSE evaluation of H-matrix estimation and the average number of features detected for both, Graf and wall, are summarized in Table 4. SURF realizes on the Graf scene on average RMSE 8.18 m and detect 1675 pt. This study's multiple FDO approach achieves 11.57 m (−3.39 m) and detects 3032 pt (+81%) points. For the Wall case, FAST realizes 6.27 m and detects 14′970 pt, whereas this study's multiple FDO approach achieves 2.62 m (+3.65 m), and detects 18,443 pt (+19%).

**Table 4.** Graf and wall average RMSE [m] and the average number of features detected.

| | AVG RMSE [m] | | Features | |
|---|---|---|---|---|
| **Column** | **1** | **2** | **3** | **4** |
| **IMG** | **Graf** | **Wall** | **Graf** | **Wall** |
| **Multi** | 11.57 | 2.62 | 3032 | 18,443 |
| **SURF** | 8.18 | 14.95 | 1675 | 3335 |
| **ORB** | 14.84 | 19.09 | 168 | 137 |
| **FAST** | 27.88 | 6.27 | 1189 | 14,970 |

In the last case, the Leuven and Ubc scenes are both structured scenes with different condition changes. The Leuven scene is a close-up with a clear fore- and background, where the illumination changes increasingly pair-by-pair. The Ubc scene is more dominantly scenery with no clear foreground with compression applied increasingly pair-by-pair.

The pairwise RMSE evaluation of H-matrix estimation is shown in Figure 10. The Leuven case shows that except for one value, all FDOs maintain an error under 1 m. The last two pairs, however, show that illumination can have a greater impact on the result. The compression rate influences the results in a constant manner. All FDOs perform under 0.3 m error. With decreasing illumination, the RMSE rises expectedly. However, the reflection of the car windows and the background make this scene much more difficult, so that the performance was a nice surprise. The JPEG compression, unexpectedly, seems to have much less influence on the RMSE.

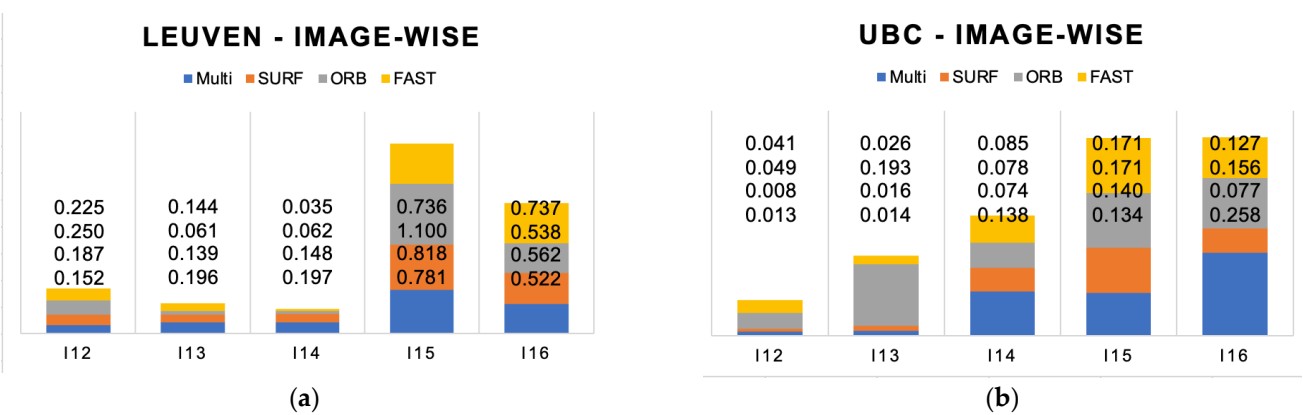

**Figure 10.** Pairwise RMSE [m] evaluation of H-matrix estimation of Leuven (**a**) and Ubc (**b**).

The average RMSE evaluation of H-matrix estimation and the average number of features detected for both, Leuven and Ubc, are summarized in Table 5. SURF and multi-FDO can both achieve 0.37 m (−+0) for the Leuven scene. However, SURF detects 3760 pt whereas this study's multi-FDO detects 10,840 pt (+188%). On the Ubc JPG compression case, SURF realizes 0.062 m and detects 3681 pt. Our multiple FDO approach achieves 0.11 m (−0.048 m), but detects 15,901 pt (+331%).

**Table 5.** Leuven and Ubc average RMSE [m] and the average number of features detected.

| | AVG RMSE [m] | | Features | |
|---|---|---|---|---|
| **Column** | **1** | **2** | **3** | **4** |
| **IMG** | **Leuven** | **Ubc** | **Leuven** | **Ubc** |
| **Multi** | 0.37 | 0.11 | 10,840 | 15,901 |
| **SURF** | 0.37 | 0.062 | 3760 | 3681 |
| **ORB** | 0.40 | 0.13 | 891 | 387 |
| **FAST** | 0.38 | 0.09 | 6189 | 11,833 |

### 5.2. EPFL Dataset

This study applies the first two images as input of the EPFL Fountain-P11 dataset. The scene consists of textured and structured components. The scene contains three major elements: first, the brick wall as a background, and the fountain as the main element in the foreground alone. It is attached to the brick wall, together with these two elements dominating the scene. The third element is a wall of homogenous texture with two windows, connected to the brick wall at 90° angle. The evaluation is divided into two parts: first, the feature accumulation (Table 6) over the individual, and in this study, the proposed multiple FDO, and second, the visible evaluation of the sparse SfM reconstruction (Figure 6).

**Table 6.** The features accumulation and contribution to the point cloud of EPFL dataset applied.

| Features | Initial | Fraction1 [%] | PCL | Fraction2 [%] | Loss | Fraction3 [%] |
|---|---|---|---|---|---|---|
| **Column** | **1** | **2** | **3** | **4** | **5** | **6** |
| **SURF** | 18,763 | 53.57 | 14,093 | 50.93 | 4670 | 24.89 |
| **ORB** | 12,980 | 36.03 | 10,266 | 37.10 | 2714 | 20.91 |
| **FAST** | 3284 | 9.38 | 2366 | 8.55 | 918 | 27.95 |
| **Multi** | 35,027 | 100 | 27,673 | 100 | 7354 | 21 |

The evaluation of the feature accumulation, summarized in Table 6, shows how the number of features changes during processing. The first and second columns display the initial absolute number of detected features and the fraction that each individual FDO contributes (fraction1 [%]). SURF detects the most features, 18,763, and contributes by far

the biggest fraction with 53.57%, whereas FAST detects the smallest number. The third and fourth columns display the final absolute number of points in the point cloud generation process and the fraction each individual FDO contributes (fraction2 [%]). SURF generates the densest point cloud; however, ORB seems to be more stable and gains a fraction. The last two columns (5 and 6) show the actual loss of points for each variation, respectively, and the absolute and the fraction (fraction3 [%]). ORB seems to be more stable while generating ~15% fewer points. FAST generates the fewest points and losses over the process ~28% of the points generated.

The 3D sparse reconstruction of all four options is displayed in Figure 11. The top row left side shows the FAST results, and the right, the ORB results. Both reconstructed the textured area of the brick wall but barely got the fountain or the second building with a window.

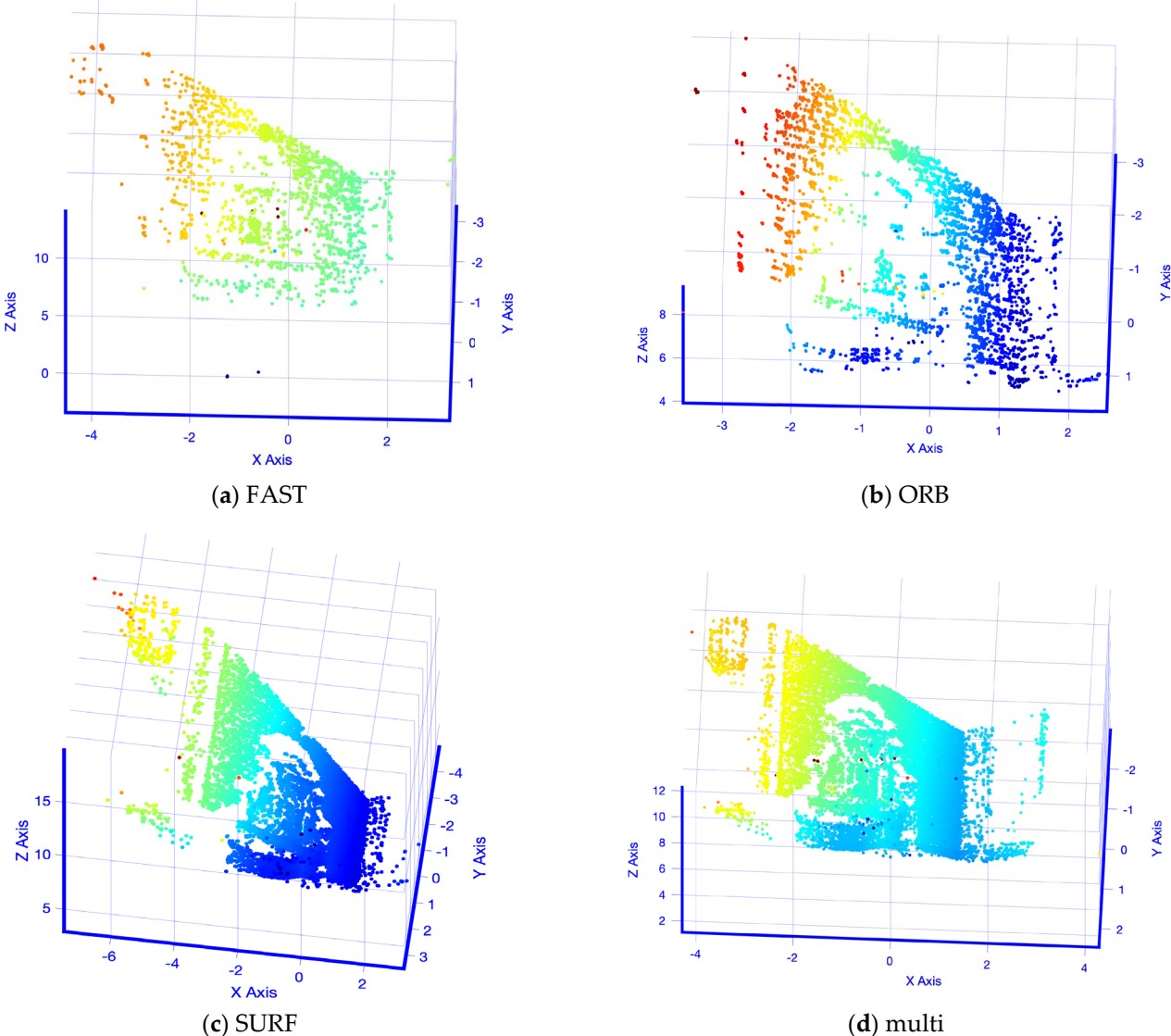

(**a**) FAST

(**b**) ORB

(**c**) SURF

(**d**) multi

**Figure 11.** Sparse Reconstruction. FAST (**a**), ORB (**b**), SURF (**c**) and this study's multi (**d**).

The bottom left displays the SURF result, which got the wall as well as the fountain recognizable. In addition, the window pattern is identifiable as well. The bottom right is the approach proposed in this study, the multiple FDO. It reconstructed the wall, as well as Fountain. Similarly to SURF, the window and the connected wall are recognizably well reconstructed.

### 5.3. CSRSR Dataset

The study applied the real-world dataset of CSRSR as introduced to reconstruct the antenna and evaluate performance. This is accomplished by the open-source software MicMac and CloudCompare. This study applied MicMac to generate a sparse point cloud to compare the result of the algorithms proposed in this study. The density analysis was made in the open-source software CloudCompare to analyze the point distribution and the location of maxima and minima.

The results of a short baseline 3D reconstruction are shown in Figure 12, in this study (a) and in MicMac (b). MicMac, as well as the approach proposed in this study, both successfully reconstructed a 3D model. The dish of the antenna and a part of the rooftop are easily recognizable. Nevertheless, the mounting tower is barely recognizable. Some input and output numbers are summarized in Table 7. This study reconstructed the antenna based on two images only and achieved 4, 378 points. MicMac achieved nearly double that amount with 7803 points. However, MicMac needs a larger number of input images and, as a direct cause, the processing time is larger as well.

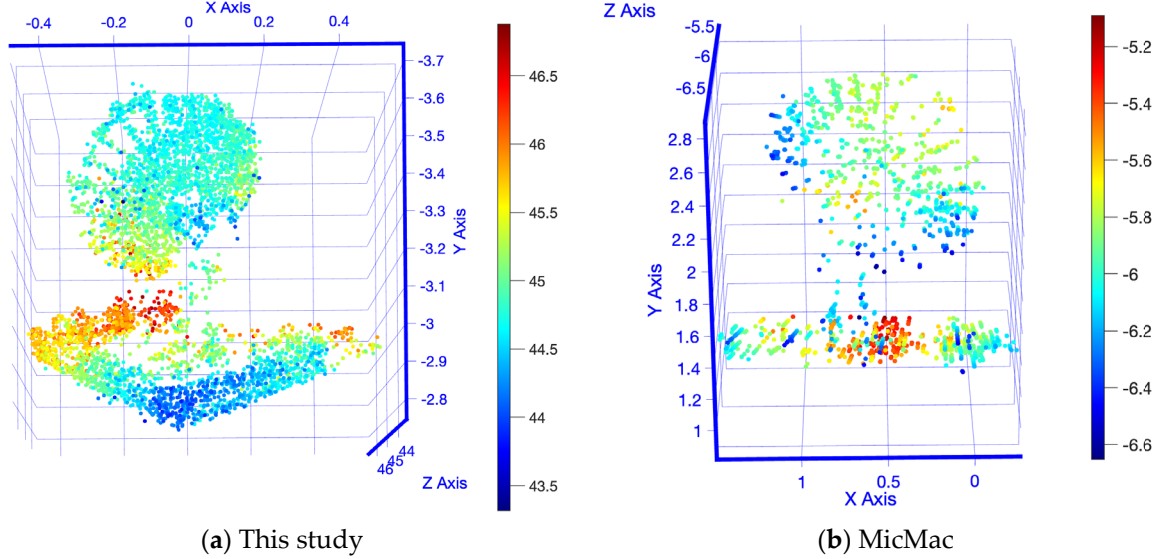

(**a**) This study                    (**b**) MicMac

**Figure 12.** Short baseline 3D reconstruction of the antenna.

**Table 7.** Input and output overview of case 1.

| Short Baseline | This Study | MicMac |
|:---:|:---:|:---:|
| **Number of Images** | 2 | 31 |
| **Initial Point Cloud** | 75,496 | 99,302 |
| **Antenna** | 4378 | 7803 |

The point density analyses are shown in Figure 13 with a histogram (Gauss) in Figure 14. The density analyses show how many neighbors one point within a radius of 0.05 m has. The result generated by MicMac achieves a higher maximum. Most values are located between 0 and 60 neighbors. In contrast, this study reaches a maximum of 50 neighbors per point. The distribution shows every four to five classes a significant local maximum. This can be validated by the plotting. The corner and edges of the roof, as well as the antenna, show areas characterized by higher point density.

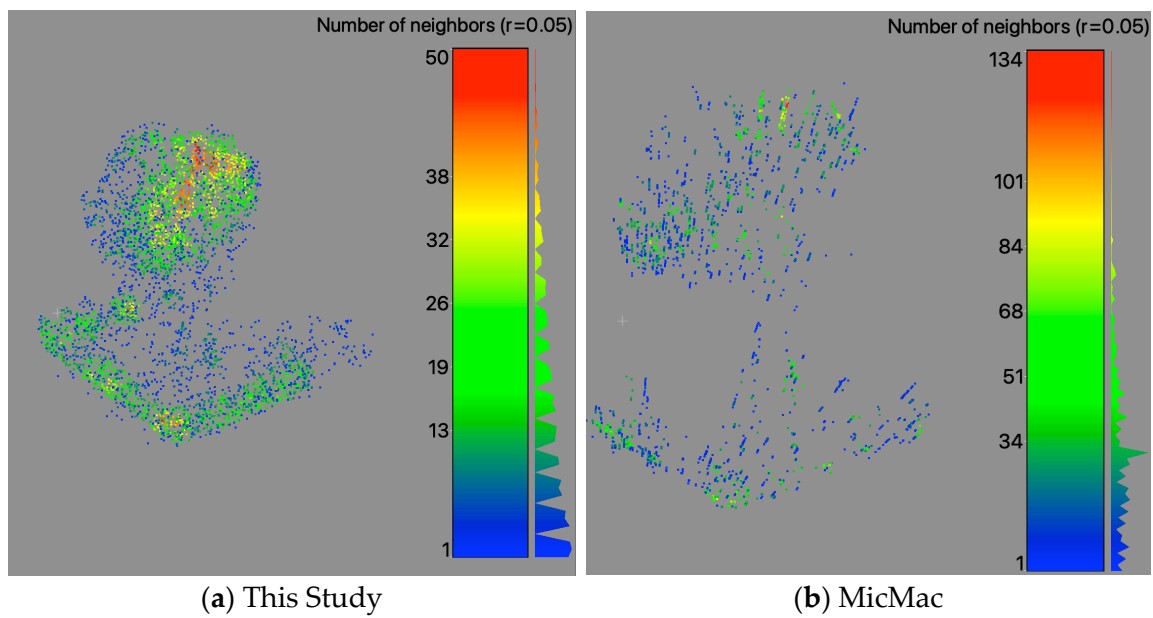

(**a**) This Study        (**b**) MicMac

**Figure 13.** Density evaluation of a small shift on the reconstruction in CloudCompare.

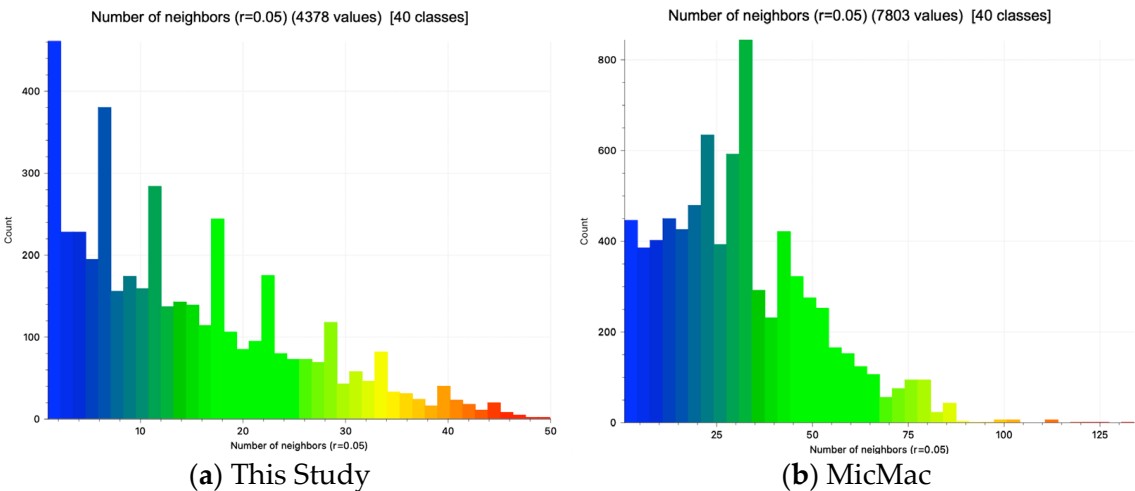

(**a**) This Study        (**b**) MicMac

**Figure 14.** Histogram of the density evaluation of a small shift on the reconstruction in CloudCompare.

The results of the 3D reconstruction shown in Figure 15 are generated with an increased baseline and camera angle. MicMac, again, generated a denser point cloud than our approach. MicMac generates a point cloud with 203,122 points, whereas our approach has 9666 points. In contrast, MicMac needs many more images (Table 8) than the approach proposed in this study and as a result, much more time. Interestingly, the reconstruction by MicMac and the approach proposed in this study is off in scale.

**Table 8.** Input and output overview of case 2.

| Long Baseline and Angle | This Study | MicMac |
| --- | --- | --- |
| Number of Images | 2 | 93 |
| Processing Time | >0.15 min | 202 min |
| Initial Point Cloud | 30,106 | 409,261 |
| Antenna | 9666 | 203,122 |

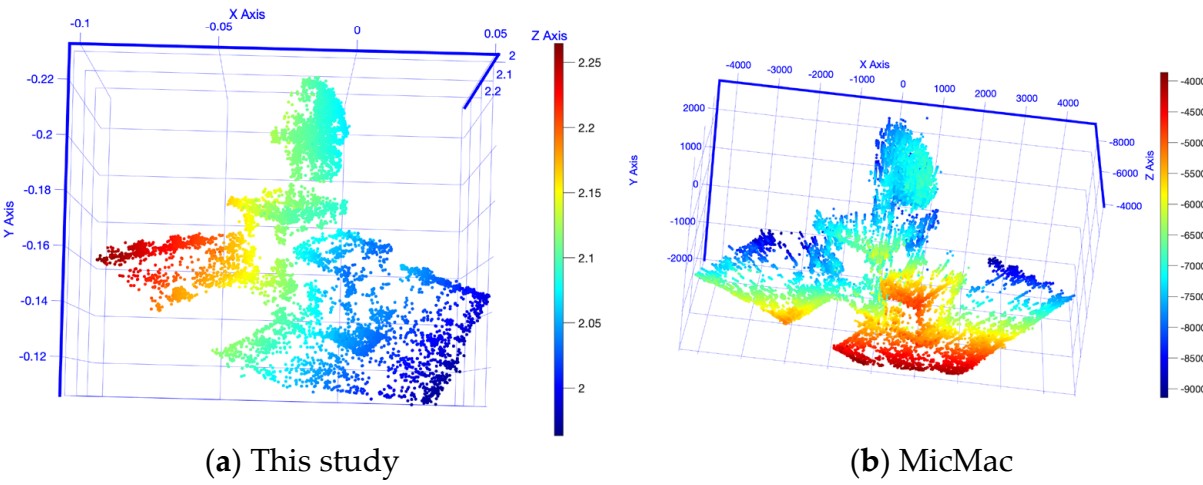

(**a**) This study   (**b**) MicMac

**Figure 15.** 3D reconstruction of the Rooftop with antenna.

This study-generated model is too small with an x-range of $-0.1$ to 0.05, a y-range of $-0.12$ to $-0.22$ and a z-range of 2.2 to 2, whereas the MicMac-generated model is too big with an x-range of $-4000$ to 4000, a y-range of $-2000$ to 2000 and a z-range of $-4000$ to $-8000$. The 3D model consisting of the antenna with mount, two auxiliary buildings and the actual rooftop are reconstructed recognizably well. The input and output are summarized in Table 8. This study generated the already-described 3D model with two images and less than 0.15 min. MicMac needed many more images and therefore needed much more time, 202 min.

The visible point density (Figure 16) is analyzed with a histogram (Gauss) (Figure 17). This study's point density radius is 0.006 m with a range of 0 to 453 divided into 99 classes (histogram). The graphic (Figure 16a) shows that the edges and corners, as well as the antenna, are locations of increased density with a mean value of 114.64 and a standard deviation of 85.86. In contrast, the MicMac result shows within a radius of 60 m with a range of 0 to 936 divided into 451 classes. The graphic (Figure 16b) shows that the edges and the corner on the right-hand side are locations of higher density with a mean value of 179.44 and a standard deviation of 114.64 (Figure 17).

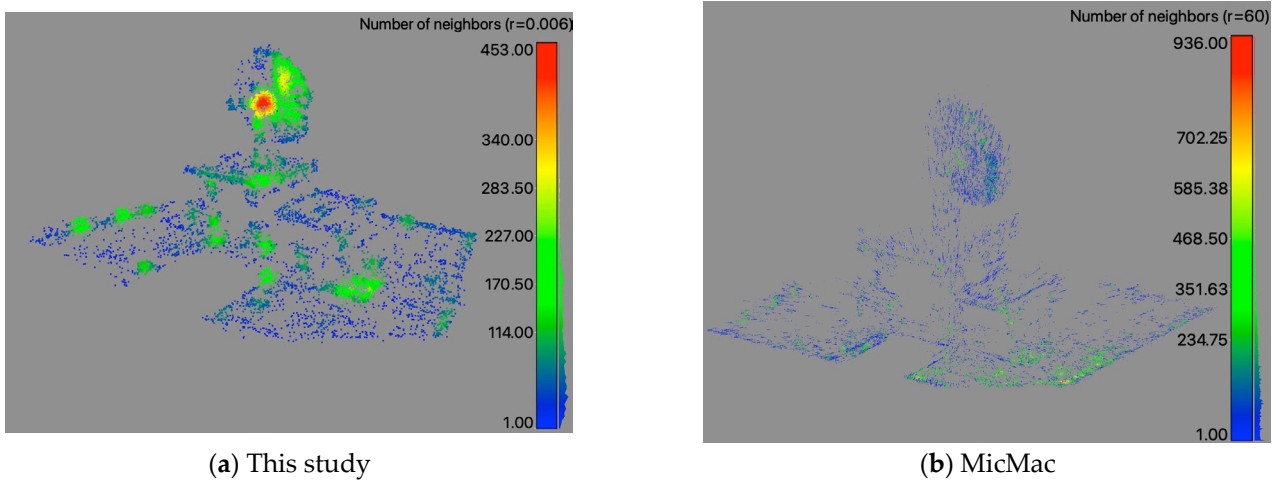

(**a**) This study   (**b**) MicMac

**Figure 16.** Density evaluation of a small shift on the reconstruction in CloudCompare.

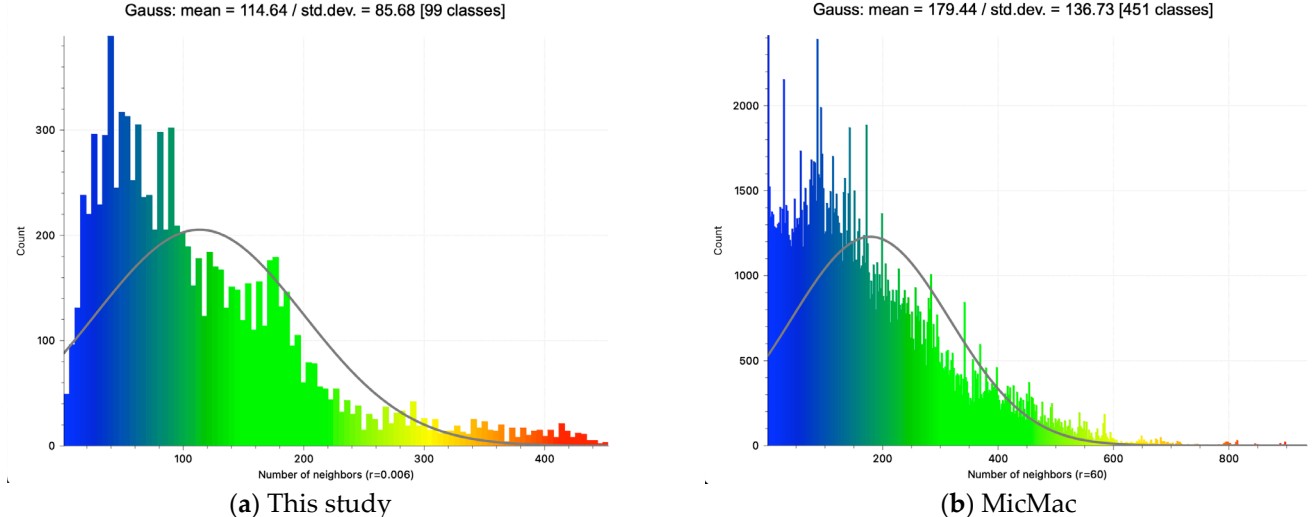

**Figure 17.** Histogram of the density evaluation of a bigger shift on the reconstruction in CloudCompare.

The influence of the combined feature detector operators is visible over the different reconstruction models and its comparison to the MicMac models is presented. The edges are locations of higher point density. The main proportion of the points is located within the first half of the spectrum. However, our study achieves a small, but clearly visible proportion within the higher point density spectrum of the histogram. The reconstruction is achieved with only two images, while MicMac needs many more images to reconstruct the point cloud.

### 6. Conclusions

The workflow proposed in this study for a two-view Structure-from-Motion algorithm has been proven to be performance-efficient and robust. Our workflow emphasizes its strength over the different condition changes of the Oxford dataset cases. The sparse 3D reconstruction based on the EPFL dataset also indicates how beneficial our approach is in contrast to a single operator. The real-world CSRSR dataset results further demonstrate the efficient and robust performance of the proposed algorithms compared with an open-source photogrammetry software in generating 3D point clouds.

Our approach of multiple FDOs achieves by applying the Oxford dataset on average RMSE 0.11 m (Ubc), 0.36 m (Bikes), 0.52 m (Trees) and 0.37 m (Leuven). The results demonstrate the strength to handle illumination changes, blurring and compression satisfying. The evaluation of the EPFL dataset focuses on the 3D reconstruction, the feature accumulation and their behavior. The point cloud generated with our approach reaches 27,673 pts, whereas ORB reaches 10,266 pts. The loss of points from matching to the final point cloud is about 21% in our approach, which is similar to ORB, 20.91% SURF. Our approach achieves the best recognizable reconstruction. The performance of applying the proposed approach on benchmark datasets such as Oxford and EPFL is promisingly good and highly efficient. The condition analyses of the Oxford dataset also indicate that our approach produced excellent results.

A third experiment on a real-world UAV dataset was performed to evaluate the performance of the proposed two-view Structure-from-Motion approach. In two cases, if we increased the baseline and viewpoint (looking angle) to further examine the workflow proposed in this study. The proposed approach generated a point cloud model with increased point cloud density around the edges, corners of the buildings and the target (antenna) by using a minimal amount of only two images. The results generated with MicMac show a larger number of points. However, MicMac needs more resources and seems to struggle with an increased baseline and angle as well. The framework proposed in

this study has proven a viable alternative to a classical procedure, in terms of performance, efficiency and simplicity.

**Author Contributions:** Conceptualization, F.T. and E.J.D.; methodology, software, validation, formal analysis, investigation, E.J.D.; resources, data curation, F.T. and E.J.D.; writing—original draft preparation, E.J.D.; writing—review and editing, F.T.; visualization, E.J.D.; supervision, project administration F.T.; All authors have read and agreed to the published version of the manuscript.

**Funding:** This research was partially supported by Ministry of Science and Technology, Taiwan (Project No.: MOST-111-2221-E-008-021-MY3).

**Institutional Review Board Statement:** Not applicable.

**Informed Consent Statement:** Not applicable.

**Data Availability Statement:** The Affine Covariant Features Datasets from Oxford University, GB Available online: https://www.robots.ox.ac.uk/~vgg/research/affine/ (accessed on 16 November 2022). The Fountain P11 dataset from École Polytechnique Fédérale de Lausanne, CH, Available online: https://documents.epfl.ch/groups/c/cv/cvlab-unit/www/data/multiview/denseMVS.html (accessed on 16 November 2022). CSRSR dataset not applicable.

**Conflicts of Interest:** The authors declare no conflict of interest.

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
