# Peer review of "Two-View Structure-from-Motion with Multiple Feature Detector Operators"

_remotesensing, doi:10.3390/rs15030605_

Round 1

Reviewer 1 Report

Thank you for the submission, see my reviews in the pdf attached.

The paper needs a huge check in the presentation of the results

Reviewer 2 Report

The paper needs to be more explicit about the primary contributions and needs to focus on their own work. A big chunk of the paper is introductions/descriptions of others’ work. The most important part, the details of how the multiple FDO is implemented, are missing. 

Reviewer 3 Report

This paper presents an approach of SfM with fusion of different feature descripton. Though that is not a novelty at all, the authors extend its application to wider PCL generation and tested over real datasets, fact that confers the work more robustness and validity to be appreciated as a contribution (at least from a practical point of view) to the scientific community.

Other comments:

- Section about Epipolar Geometry definitions and theory, as well as, feature detectors essentials are mainstream and very outdated (the explanation itself). Please sum up and simply cite.

- Please revise in depth the use of acronyms with capital letters, and introduced from the very fist time.

- Also, double check the use of numbers in the text, when it looks formal by texting the number (ie "divided in to 2" looks very unformal, line 355, page 9).

Round 2

Reviewer 1 Report

I think the authors have reviewed the paper accordingly with the suggestins
